# Evaluating the efficacy and safety of mavacamten in hypertrophic cardiomyopathy: A systematic review and meta-analysis focusing on qualitative assessment, biomarkers, and cardiac imaging

Rahul Vyas[1], Viraj Panchal[2], Shubhika Jain[3]*, Manush Sondhi[1], Mansunderbir Singh[1], Keerthish Jaisingh[4], Sahith Reddy Thotamgari[4], Anuj Thakre[1], Kalgi Modi[4]

1 Department of Internal Medicine, Louisiana State University, Shreveport, Louisiana, United States of America, 2 Department of Medicine, Smt. NHL Municipal Medical College and SVPISMR, Ahmedabad, Gujarat, India, 3 Department of Medicine, Kasturba Medical College, Manipal, Karnataka, India, 4 Department of Cardiology, Louisiana State University, Shreveport, Louisiana, United States of America

☯ These authors contributed equally to this work.
* shubhikajn24@gmail.com

## Abstract

### Background

Hypertrophic Cardiomyopathy (HCM) is a complex cardiac condition characterized by hypercontractility of cardiac muscle leading to a dynamic obstruction of left ventricular outlet tract (LVOT). Mavacamten, a first-in-class cardiac myosin inhibitor, is increasingly being studied in randomized controlled trials. In this meta-analysis, we aimed to analyse the efficacy and safety profile of Mavacamten compared to placebo in patients of HCM.

### Method

We carried out a comprehensive search in PubMed, Cochrane, and clinicaltrials.gov to analyze the efficacy and safety of mavacamten compared to placebo from 2010 to 2023. To calculate pooled odds ratio (OR) or risk ratio (RR) at 95% confidence interval (CI), the Mantel-Haenszel formula with random effect was used and Generic Inverse Variance method assessed pooled mean difference value at a 95% CI. RevMan was used for analysis. P<0.05 was considered significant.

### Results

We analyzed five phase 3 RCTs including 609 patients to compare mavacamten with a placebo. New York Heart Association (NYHA) grade improvement and KCCQ score showed the odds ratio as 4.94 and 7.93 with p<0.00001 at random effect, respectively. Cardiac imaging which included LAVI, LVOT at rest, LVOT post valsalva, LVOT post-exercise, and reduction in LVEF showed the pooled mean differences for change as -5.29, -49.72, -57.45,

**Data Availability Statement:** All relevant data are within the paper and its Supporting Information files.

**Funding:** The author(s) received no specific funding for this work.

**Competing interests:** The authors have declared that no competing interests exist.

-36.11, and -3.00 respectively. Changes in LVEDV and LVMI were not statistically significant. The pooled mean difference for change in NT-proBNP and Cardiac troponin-I showed 0.20 and 0.57 with p<0.00001. The efficacy was evaluated in 1) A composite score, which was defined as either 1·5 mL/kg per min or greater increase in peak oxygen consumption (pVO2) and at least one NYHA class reduction, or a 3·0 mL/kg per min or greater pVO2 increase without NYHA class worsening and 2) changes in pVO2, which was not statistically significant. Similarly, any treatment-associated emergent adverse effects (TEAE), treatment-associated serious adverse effects (TSAE), and cardiac-related adverse effects were not statistically significant.

## Conclusion

Mavacamten influences diverse facets of HCM comprehensively. Notably, our study delved into the drug's impact on the heart's structural and functional aspects, providing insights that complement prior findings. Further large-scale trials are needed to evaluate the safety profile of Mavacamten.

## 1. Introduction

Hypertrophic Cardiomyopathy (HCM) is a complex cardiac condition characterized by diverse etiological factors, leading to a spectrum of clinical manifestations [1]. This condition encompasses both familial and acquired forms, rooted in genetic mutations affecting sarcomere proteins. Histopathological changes, including myocyte disarray, hypertrophy, and fibrosis, result in asymmetrical left ventricular hypertrophy and a range of distressing symptoms. Most diagnoses occur after symptom onset and significant myocardial remodeling, primarily due to the rarity of the disease. Current treatments mainly focus on symptom relief, leaving the underlying causes unaddressed [2]. In this context, our meta-analysis centers on the promising novel drug, Mavacamten, which targets the pathophysiological mechanisms of HCM [3].

In our comprehensive analysis, we have categorized the results into crucial domains, including baseline characteristics, qualitative assessments, cardiac imaging, biomarkers, clinical parameters, and safety profiles, to elucidate the efficacy and safety of Mavacamten in HCM.

## 2. Methods

We conducted a systematic review and a meta-analysis to report the efficacy and safety of mavacamten compared to placebo in hypertrophic cardiomyopathy patients according to the Preferred Reporting Items for Systematic Reviews and Meta-analyses (PRISMA) guidelines [4]. The PRISMA checklist is shown in S1 Fig.

### 2.1 Endpoints

The objective of this analysis was to evaluate efficacy of mavacemten compared to placebo in HCM patients in terms of a (1) composite scoring which showed either 1·5 mL/kg per min or greater increase in peak oxygen consumption (pVO2) and at least one New York Heart Association (NYHA) class reduction, or a 3·0 mL/kg per min or greater pVO2 increase without NYHA class worsening compared (2) Change in pVO2 (3) improvement in NYHA class from baseline.

Changes in echocardiography parameters were evaluated to report significant changes in the imaging findings in patients taking mavacamten compared to placebo focusing on (1) change in Left Ventricular Outflow Tract (LVOT) gradient at rest, after valsalva, and post-exercise (2) reduction in Left Ventricular Ejection Fraction (LVEF%) (3) change in Left Atrial Volume Index (LAVI) (mL/m$^2$) from baseline, and (4) Left Ventricular Mass Index (LVMI) from baseline in HCM patients. Furthermore, a qualitative improvement score reporting a change in the Kansas City Cardiomyopathy Questionnaire in a Clinical Summary Score (KCCQ-CSS) was used to evaluate the association between mavacamten and placebo in HCM patients.

N-terminal pro–B-type natriuretic peptide (NT-proBNP) and High-sensitivity cardiac troponin I (hs-cTnI) proportionate mean reduction was evaluated to measure treatment-related changes in biomarkers among the mavacamten compared to placebo in HCM patients. Lastly, more than one treatment-related serious, emergent, and cardiac adverse events (which included atrial fibrillation, sudden cardiac death, congestive heart failure, and coronary artery disease) were evaluated as a safety outcome in patients taking mavacamten compared to placebo.

## 2.2 Search strategy and selection criteria

We carried out a comprehensive search in PubMed, Cochrane, and clinicaltrials.gov to analyze the efficacy and safety of mavacamten compared to placebo using the following MeSH terms: "mavacamten" or "camzyos" and "hypertrophic cardiomyopathy" or "oHCM" or "HCM" or "HOCM" from 2010 to 2023. Relevant randomized controlled trials (RCTs), or any prospective study that focused on objectives were included in primary screening. After removing duplicates, we excluded review articles, any qualitative data or difficult-to-interpret values either shown in graphs or any figures, or any non-English texts during secondary screening. If a study was a subanalysis of an RCT, then the RCT was considered and not the subanalysis to remove any data duplication. However, if any subanalysis provided extra information that was not previously mentioned in the primary RCT then we considered to include that data for analysing any particular parameter. Similarly, if a study had provided values for Intention To Treat (ITT) as well as Per Protocol Analysis (PPA) analysis, then values corresponding to ITT analysis were taken into consideration to avoid any bias. Two investigators (SJ, VP) examined each study for any potential flaws, and any disagreement was settled through discussion.

## 2.3 Data collection

Included studies were evaluated to summarize the author's name, publication year, number of participants in the mavacamten and placebo group, as well as the relevant information on endpoints. Among the selected studies, those who failed to provide the outcome in mean difference were still included after calculating the value using the online calculator MedCalc [5] from the data provided in the original study. Studies that compared mavacamten with a placebo were only taken into consideration irrespective of the background therapy, which included beta-blockers or calcium channel blockers.

## 2.4 Assessment of risk of bias

The risk of bias for the included RCTs was assessed by two independent reviews (VP, SJ) following the guidelines in the Cochrane Handbook for Systematic Reviews of Interventions [6]. Various parameters such as selection bias, allocation bias, blinding, incomplete reporting of data, and any other type of bias were evaluated and graded as low, unclear, or high risk, as shown in the S2 Fig.

## 2.5 Statistical analysis

We compared mavacamten with placebo to calculate pooled odds ratio (OR) or risk ratio (RR) at 95% confidence interval (CI) for dichotomous data. The Mantel-Haenzel formula with random effect was used to create forest plots. An OR/RR of 1 means that the new treatment and the placebo have equivalent effects. If improvement is associated with higher scores on the outcome measure, an OR/RR greater than 1 indicates the degree to which treatment showed more effect than a placebo, and an OR/RR less than 1 indicates the degree to which treatment is less efficacious than a placebo. The generic Inverse Variance method was used to analyze the pooled mean difference value of mavacamten compared to placebo at 95% CI to create forest plots. Point estimates in terms of overall mean difference were used, while the data entered was calculated on a log scale and standard errors (SE) were combined by the Generic Inverse Variance method.

Meta-analysis was performed using RevMan version 5.4. The random effect model was preferred over the fixed effect model to avoid the likelihood of interstudy variability affecting the effect estimate. Publication biases were also evaluated using funnel plots and forest plots. Statistical Heterogeneity (I2) which demonstrated the variation among the included studies was used to evaluate the heterogeneity. The I2 statistic of >75% was considered significant heterogeneity, and if reported, a sensitive analysis using a "leave-out" analysis technique in random effect was carried out. P value <0.05 was considered to be statistically significant.

## 3. Results

### 3.1 Baseline characteristics

In the primary screening, 156 articles were obtained through searching the database using the MeSH words. Out of those, 147 were excluded for being systematic reviews or failing to provide sufficient data for analysis. A total of 9 articles were included for a secondary screening for a full-text review based on the title and abstract details. Out of those 9, two studies were excluded as they were the extension studies of the primary RCTs, and another two studies compared mavacamten with the combination of mavacamten and beta blockers instead of placebo [7–10]. Lastly, five phase 3 randomized controlled trials were included in our analysis comparing mavacamten with placebo on a background of either beta-blockers or calcium channel blockers, as shown in Fig 1 [11–15].

### 3.2 Qualitative assessment parameters

The Kansas City Cardiomyopathy Questionnaire-Clinical Summary Score (KCCQ-CSS) and NYHA grading were included as the parameters to assess quality improvement in both Mavacamten and placebo groups. The data from 4 studies were included to analyze the results.

For NYHA grade improvement, the odds ratio was 3.55 with a 95% confidence interval (CI) of 1.47 to 8.56. However, 76% heterogeneity was noted at a p-value of 0.006 at the random effect (Fig 2).

We performed a sensitive analysis excluding the study by Ho et al. giving us a heterogeneity of 0%. In the sensitive analysis, the odds ratio increased to 4.94 with a 95% CI of 3.25 to 7.49 which was significant at random effect with a p< 0.00001 (Fig 3).

For KCCQ-CSS, the mean difference was 7.93 with a 95% CI of 4.52 to 11.35 which was significant with p<0.00001 at random effect (Fig 4). There was 47% heterogeneity seen which did not meet the criteria for high heterogeneity and hence sensitive analysis was not performed.

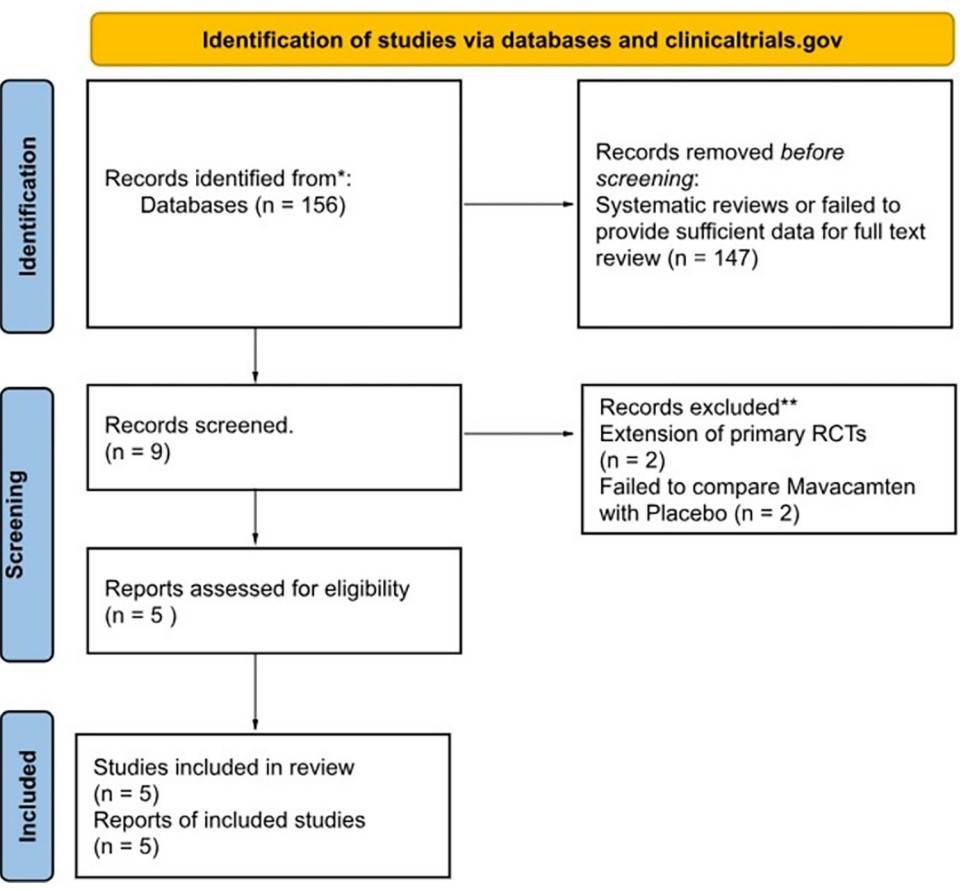

**Fig 1. PRISMA flow diagram of the included studies in the meta-analysis.**

### 3.3 Cardiac imaging parameters

Echocardiographic findings including change in LVEDV, LAVI, LVMI, and LVOT gradient (at rest, post valsalva, and post-exercise), and LVEF were analyzed to assess improvement in both the Mavacamten and placebo group.

The data from 2 studies were analyzed to assess the change in LVEDV. The mean difference of -1.36 with a 95% confidence interval of -4.35 to 1.62 was noted at a p-value of 0.37 which was not significant.

The data from 3 studies were analyzed to assess the changes in LAVI. The mean difference of -5.29 with a 95% confidence interval of -8.20 to—2.37 was noted which was significant with a p-value of 0.0004 at random effect (Fig 5). A heterogeneity of 65% was noted which did not meet the criteria for high heterogeneity and hence sensitive analysis was not performed.

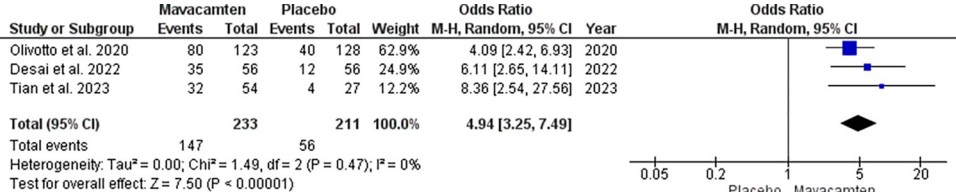

**Fig 2. Forest plot showing NYHA grade improvement in the Mavacamten group compared to placebo at random effect.**

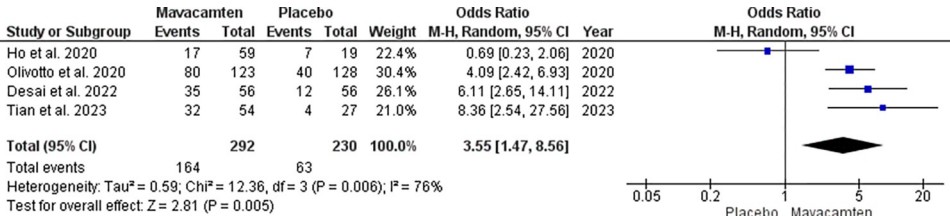

**Fig 3. Leave-out analysis with the forest plot showing NYHA grade improvement in the Mavacamten group compared to placebo at random effect.**

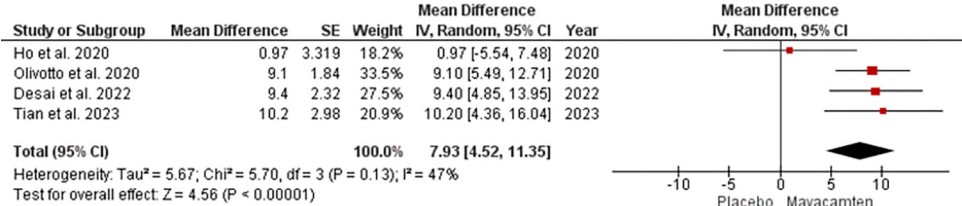

**Fig 4. Forest plot showing KCCQ-CSS score of the Mavacamten group compared to placebo at random effect.**

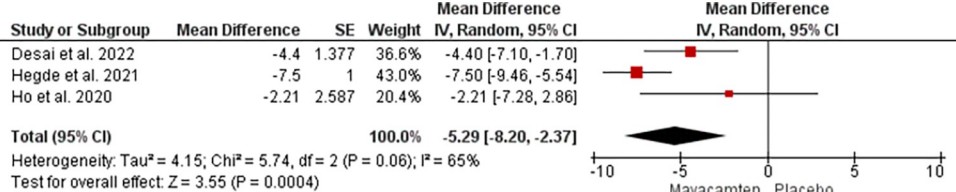

**Fig 5. Forest plot showing changes in LAVI of the Mavacamten group compared to placebo at random effect.**

The data from 3 studies were analyzed to assess the change in LVMI. The mean difference of -6.58 with a 95% confidence interval of -31.22 to 18.06 was noted at a p-value of 0.60 which was not significant.

The data from 2 studies were analyzed to assess changes in LVOT at rest, post-valsalva, and post-exercise with mean differences of -49.72 (-67.91 to -31.54), -57.45 (-79.50 to -35.40) and -36.11 (-42.33 to -29.90) respectively with p<0.00001 at random effect with insignificant heterogeneity (Figs 6–8).

The data from 3 studies were analyzed to assess reduction in LVEF. The mean difference of -3.00 with a 95% CI of -5.16 to -0.83 was noted at a p-value of 0.007 at random effect (Fig 9).

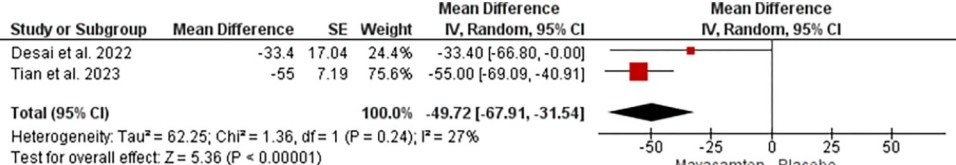

**Fig 6. Forest plot showing changes in LVOT at rest of the Mavacamten group compared to placebo at random effect.**

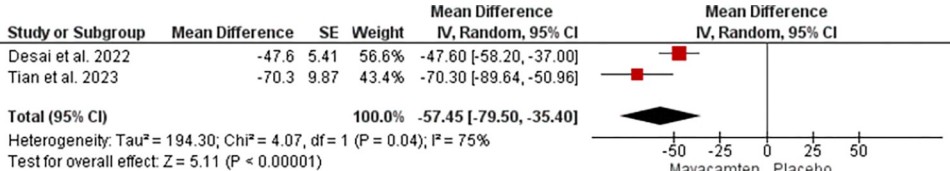

**Fig 7. Forest plot showing changes in LVOT at post-valsalva of the Mavacamten group compared to placebo at random effect.**

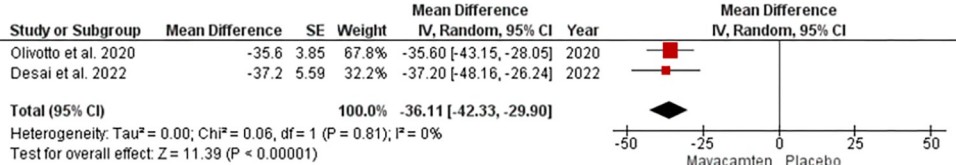

**Fig 8. Forest plot showing changes in LVOT at post-exercise of Mavacamten compared to placebo at random effect.**

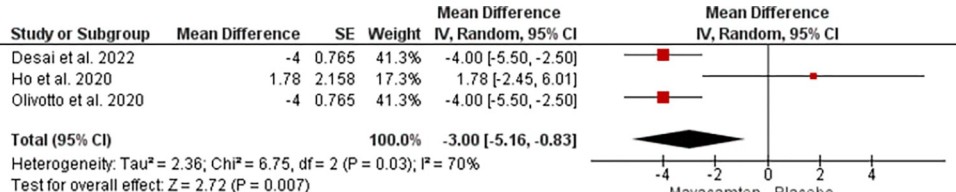

**Fig 9. Forest plot showing a reduction in LVEF of Mavacamten compared to placebo at random effect.**

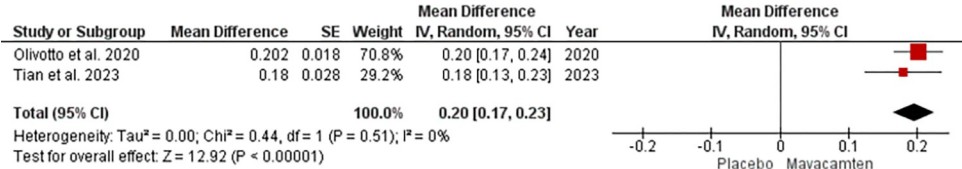

**Fig 10. Forest plot showing changes in NT-proBNP of Mavacamten group compared to placebo at random effect.**

## 3.4 Biomarkers measurement

Changes in NT-proBNP and Cardiac troponin-I were analyzed to assess improvement in both the Mavacamten and placebo groups. The data from 3 studies were included to analyze the results.

For NT-proBNP changes, the mean difference of 0.23 with a 95% CI of 0.16 to 0.30 was noted. However, the heterogeneity of 80% was noted at a p-value of 0.007 (Fig 10).

We performed a sensitivity analysis by excluding the study by Desai et al. giving us the heterogeneity of 0%. In that sensitivity analysis, the mean difference decreased to 0.20 with a 95% CIl of 0.17 to 0.23 with p<0.00001 (Fig 11).

For Cardiac troponin-I, the mean difference of 0.48 with a 95% CI of 0.31 to 0.66 was noted. However, the heterogeneity of 88% was noted at a p-value of 0.0002 (Fig 12).

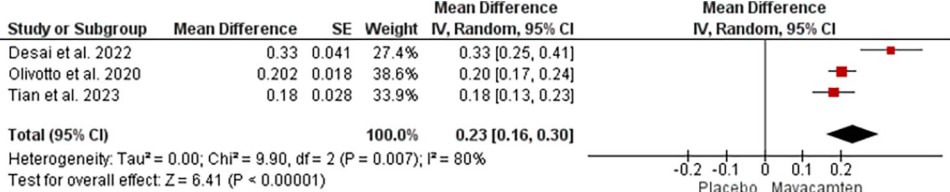

**Fig 11. Leave-out analysis with the forest plot showing changes in NT-proBNP of the Mavacamten group compared to placebo at random effect.**

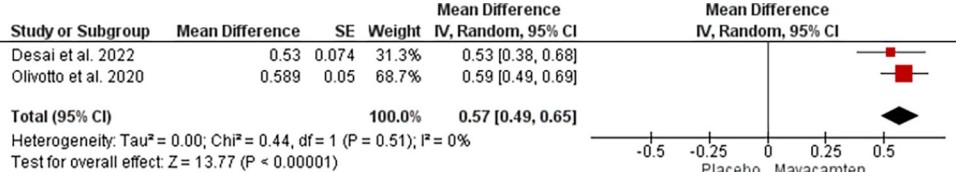

**Fig 12. Forest plot showing changes in Cardiac troponin-I of the Mavacamten group compared to placebo at random effect.**

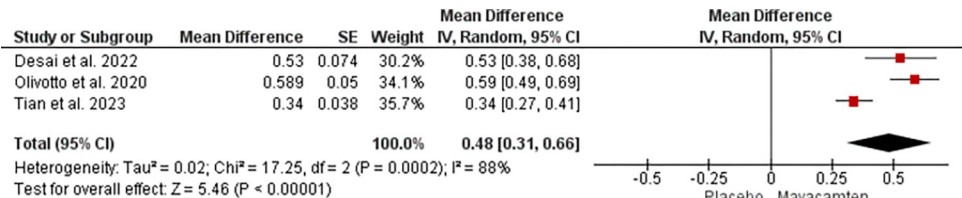

**Fig 13. Leave-out analysis with the forest plot showing changes in Cardiac troponin-I of the Mavacamten group compared to placebo at random effect.**

We performed a sensitive analysis by excluding the study by Tian et al. giving us the heterogeneity of 0%. In that sensitive analysis, the mean difference decreased to 0.57 with a 95% CI of 0.49 to 0.65 with p<0.00001 (Fig 13).

### 3.5 Clinical parameters

The data from 2 studies were included to analyze the clinical parameters of composite score and changes in pVO2.

For composite score improvement, the odds ratio was 2.12 with a 95% CI of 0.92 to 4.88 which was not significant with a p-value of 0.08 at random effect. For changes in pVO2, the mean difference was 1.18 with a 95% CI of 0.35 to 2.01 which was significant at a p-value of 0.005 at random effect.

### 3.6 Safety profile

The data from 4 studies were analyzed to assess treatment-associated emergent adverse effects (TEAE), treatment-associated serious adverse effects (TSAE), and cardiac-related adverse effects.

The measured risk ratios for TEAE, TSAE, and cardiac-related adverse effects were 1.09 (0.97 to 1.24), 0.97(0.47 to 1.97), and 0.96(0.61 to 1.51) respectively which was not significant with p>0.05 at random effect.

## 4. Discussion

The pathophysiological abnormality in HCM is the hypercontractility of the cardiac muscle leading to a dynamic obstruction of the LVOT. Mavacamten, a first-in-class cardiac myosin inhibitor, is increasingly being studied in randomized controlled trials. In this meta-analysis, we summarize the results of five phase 3 clinical trials into several key domains, including qualitative assessment parameters, cardiac imaging parameters, biomarker measurements, clinical parameters, and the safety profile of Mavacamten compared to placebo in patients of hypertrophic cardiomyopathy.

HCM is predominantly being diagnosed in the younger population, so it often becomes imperative to provide a treatment that leads to improvement in quality of life [16]. In our meta-analysis, a significant increase in KCCQ-CSS and an improvement in NYHA grading was observed in the mavacemten group compared to placebo. An improved NYHA class which is an objective score adds value to the quality assessment parameters supporting the use of mavacamten in HCM over placebo [17]. On the other hand, KCCQ being a subjective scoring is recommended to quantify the quality of care to improve patient-centered care. While KCCQ measures the patient's symptoms, it also takes into consideration physical and social limitations interfering with the quality of life in patients with heart failure [18,19]. Furthermore, the KCCQ score has been able to demonstrate prognostic significance as an association from the time of discharge for acutely decompensated heart failure [20], 1 week after hospital discharge [21], and in outpatient settings as well [22]. One of the reasons for the independent association of KCCQ scores with clinical events may be due to a domain that is otherwise not taken into consideration of other clinical markers of disease severity, such as ejection fraction or B-type natriuretic peptide [23]. When comparing 2 patients with similar estimated risk of clinical events, a patient with a lower KCCQ score is at a higher chance of experiencing death or hospitalization than a patient with similar clinical risk with a higher KCCQ score [20]. In our meta-analysis, the increased KCCQ score associated with mavacamten supports the evidence to improve the quality of care through medications in HCM patients. It enables physicians to incorporate a prognostic score routinely.

RCTs included in this analysis have also looked into echocardiographic parameters to quantify and compare the improvement in HCM symptoms between the two groups. LAVI is a measure of LA filling pressure that can be attributed to elevated LVOT gradient, mitral regurgitation due to systolic anterior motion of the mitral valve, and diastolic dysfunction associated with worsening obstructive HCM [24]. Greater baseline LVOT gradient is associated with a greater reduction in LAVI post-treatment, as a higher LVOT gradient is associated with a greater increase in LA size due to mitral regurgitation [14]. In a study by Hegde et al., LAVI was the only imaging parameter that was associated with functional status improvement, compared with other parameters, including LVOT gradient reduction. Two-dimensional cardiac imaging parameters like LVMI, with their limitations, have also shown significant improvement in the mavacamten group compared to placebo, particularly in asymmetric hypertrophy. Reduction in LVOT or afterload, or fewer actin-myosin cross-bridges, can also reflect the associated LVMI changes post mavacamten treatment compared to placebo [25–27]. Reduction in LVOT gradient at rest, post-valsalva, and post-exercise were assessed in most clinical trials to test the efficacy of mavacamten in releasing obstructive HCM as a non-invasive intervention compared to septal reduction therapy [11]. Patients with LVOT obstruction in HCM advance more rapidly to higher NYHA grade with an annual rate of 3.2%-7.4% vs 1.6% [28–30]. Hence, LVOT reduction is fundamental in oHCM treatment. LVEF in most trials was measured as a safety parameter to titrate the dose of mavacamten. In all the trials included in this meta-analysis, reversible reduction in LVEF was noted, which improved with

temporary treatment interruption and hence was not included as a side effect. However, further studies are required to assess this on a larger scale further [11,13].

One of the added benefits of Mavacamten being a direct myosin inhibitor is that it addresses the pathophysiological abnormality of HCM. The significant changes in lowering the circulating cardiac biomarkers, namely NT-proBNP and Troponin-I, further support improved long-term outcomes in HCM by mavacamten compared to placebo [31]. A significant reduction in both NT-proBNP and Troponin-I was noted in our meta-analysis. Likewise, a decrease in cardiac biomarkers was also reported in a study done by Ho et al. [13] conducted on non-obstructive hypertrophic cardiomyopathy patients. Reducing LVOT obstruction or the lusitropic effect of mavacamten by increasing the relaxation time in cardiac myosin may be the reason for lowering the cardiac biomarkers [11]. To understand these effects and the role of mavacamten in reducing cardiac biomarkers, more investigation is still needed [32].

In all the RCTs we included in our analysis, only mild-to-moderate treatment-associated-emergent adverse effects, treatment-associated-serious adverse effects, and cardiac-related adverse effects were reported with mavacamten groups. However, on comparing the events, the difference in the safety profile of both groups was not statistically significant. Hence, further long-term studies are required to evaluate the safety profile of Mavacamten.

## 4.1 Conclusion

Our study presents a unique analysis of phase 3 randomized controlled trials investigating Mavacamten's effects on HCM patients. Our approach goes beyond conventional efficacy and safety evaluations. We meticulously examined cardiac imaging parameters (e.g., left atrial volume index, left end-diastolic volume), biomarkers (NT-proBNP, cardiac troponin I), and clinical parameters to understand how Mavacamten influences diverse facets of HCM comprehensively. Notably, our study delved into the drug's impact on the heart's structural and functional aspects, providing insights that complement prior findings. Our emphasis on safety comparisons between treated and placebo groups underscores the necessity of assessing Mavacamten's risk-benefit profile in HCM treatment contexts [33,34]. While previously published data have shown the efficacy of mavacamten in HCM in terms of improvement in NYHA grading, biomarker changes and treatment related adverse events however they did not consider the clinical parameters used to evaluate the outcome [35,36]. This meta-analysis is unique in sense that it also focused on the clinical improvement such as improvement in composite score as well as change in pVO2. Nevertheless, further research, particularly through larger clinical trials, is imperative to strengthen the evidence supporting the role of Mevacamten in the treatment of HCM.

## 4.2 Study limitation

One of the limitations of our meta-analysis was the inherent heterogeneity bias among the included studies. We conducted a sensitive analysis by excluding the study that showed maximum bias. However, the confounding factor was not addressed. The included studies had different periods to measure the primary endpoint, which might be why. However, the odds ratio was considered, which does avoid time as a factor. By conducting the meta-analysis of five phase 3 clinical trials, we increased the power of this study to support the estimate of effect size obtained and decrease the variation. This is the first known meta-analysis considering various parameters to support clinical decisions. Further research is needed by conducting more randomized observational multicenter studies to support the results obtained through this analysis.

## Supporting information

**S1 Fig. PRISMA checklist of the meta-analysis.**
(DOCX)

**S2 Fig. Risk of bias of included studies in the meta-analysis.**
(DOCX)

**S1 Appendix. Dataset use to draw results and conclusions.**
(DOCX)

## Author Contributions

**Conceptualization:** Rahul Vyas, Shubhika Jain, Manush Sondhi, Mansunderbir Singh, Keerthish Jaisingh, Sahith Reddy Thotamgari, Anuj Thakre, Kalgi Modi.

**Data curation:** Rahul Vyas, Viraj Panchal, Shubhika Jain, Manush Sondhi, Mansunderbir Singh, Keerthish Jaisingh, Sahith Reddy Thotamgari, Anuj Thakre, Kalgi Modi.

**Formal analysis:** Rahul Vyas, Viraj Panchal, Shubhika Jain, Manush Sondhi, Mansunderbir Singh, Keerthish Jaisingh, Sahith Reddy Thotamgari, Anuj Thakre, Kalgi Modi.

**Investigation:** Viraj Panchal.

**Methodology:** Rahul Vyas, Viraj Panchal, Shubhika Jain, Manush Sondhi, Mansunderbir Singh, Keerthish Jaisingh, Sahith Reddy Thotamgari, Anuj Thakre, Kalgi Modi.

**Project administration:** Rahul Vyas, Viraj Panchal, Shubhika Jain, Manush Sondhi, Mansunderbir Singh, Keerthish Jaisingh, Sahith Reddy Thotamgari, Anuj Thakre, Kalgi Modi.

**Resources:** Rahul Vyas, Viraj Panchal, Shubhika Jain, Manush Sondhi, Mansunderbir Singh, Keerthish Jaisingh, Sahith Reddy Thotamgari, Anuj Thakre, Kalgi Modi.

**Software:** Rahul Vyas, Viraj Panchal, Shubhika Jain, Manush Sondhi, Mansunderbir Singh, Keerthish Jaisingh, Sahith Reddy Thotamgari, Anuj Thakre, Kalgi Modi.

**Supervision:** Rahul Vyas, Viraj Panchal, Shubhika Jain, Manush Sondhi, Mansunderbir Singh, Keerthish Jaisingh, Sahith Reddy Thotamgari, Anuj Thakre, Kalgi Modi.

**Validation:** Rahul Vyas, Viraj Panchal, Shubhika Jain, Manush Sondhi, Mansunderbir Singh, Keerthish Jaisingh, Sahith Reddy Thotamgari, Anuj Thakre, Kalgi Modi.

**Visualization:** Rahul Vyas, Viraj Panchal, Shubhika Jain, Manush Sondhi, Mansunderbir Singh, Keerthish Jaisingh, Sahith Reddy Thotamgari, Anuj Thakre, Kalgi Modi.

**Writing – original draft:** Rahul Vyas, Viraj Panchal, Shubhika Jain, Manush Sondhi, Mansunderbir Singh, Keerthish Jaisingh, Sahith Reddy Thotamgari, Anuj Thakre, Kalgi Modi.

**Writing – review & editing:** Rahul Vyas, Viraj Panchal, Shubhika Jain, Manush Sondhi, Mansunderbir Singh, Keerthish Jaisingh, Sahith Reddy Thotamgari, Anuj Thakre, Kalgi Modi.

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
