## [Decision Letter · Decision Letter 0]

10 Jan 2024

PONE-D-23-42379Evaluating the efficacy and safety of mavacamten in hypertrophic cardiomyopathy: A systematic review and meta-analysis focusing on qualitative assessment, biomarkers, and cardiac imaging.PLOS ONE

Dear Dr. Jain,

Thank you for submitting your manuscript to PLOS ONE. After careful consideration, we feel that it has merit but does not fully meet PLOS ONE’s publication criteria as it currently stands. Therefore, we invite you to submit a revised version of the manuscript that addresses the points raised during the review process.

We look forward to receiving your revised manuscript.

Kind regards,

Alberto Aimo, MD

Academic Editor

PLOS ONE

Journal Requirements:

Reviewers' comments:

Reviewer's Responses to Questions

**Comments to the Author**

1. Is the manuscript technically sound, and do the data support the conclusions?

Reviewer #1: Yes

Reviewer #2: Yes

2. Has the statistical analysis been performed appropriately and rigorously? 

Reviewer #1: Yes

Reviewer #2: Yes

3. Have the authors made all data underlying the findings in their manuscript fully available?

Reviewer #1: Yes

Reviewer #2: Yes

4. Is the manuscript presented in an intelligible fashion and written in standard English?

Reviewer #1: Yes

Reviewer #2: Yes

5. Review Comments to the Author

Reviewer #1: A nicely presented review with relevant RCTs and trials for mavacamtem in the setting of HOCM. I'would suggest adding speckle tracking imaging data, where and if available (GLS%), to echocardiographic parameters. Furthermore, I would add a section specific fo CMR data.

Reviewer #2: This is a meta-analysis of phase 3 randomized controlled trials (RCTs) investigating mavacamten's effects on hypertrophic cardiomyopathy compared to placebo. Despite the existence of at least five previous meta-analyses on this topic in the scientific literature, this meta-analysis is purportedly the most comprehensive one performed to date.

Major issues to address:

• The claim of including five phase 3 RCTs on mavacamten and excluding all subanalyses of RCTs is contradicted by the inclusion of the study by Hedge et al., which is a subanalysis of EXPLORER-HCM (Olivotto et al.). While the inclusion of the Hedge et al. subanalysis is considered reasonable due to its unique information on echocardiographic data, the claim of encompassing five RCTs should be thoroughly reviewed throughout the text.

• In the discussion, it is recommended to cite the other two recently published meta-analyses on mavacamten which include the same four trials as the present study (DOI: 10.1007/s10741-023-10375-6; DOI: 10.1186/s43044-023-00427-5). Emphasis should be placed on elucidating the novelty of the current meta-analysis compared to these prior studies.

Minor issues to address:

• Replace "pro-BNP" with "NT-proBNP."

• Ensure that acronyms are spelled out in their extended form the first time they are cited in the text (e.g., "NT-proBNP", "hs-cTnI", and so forth).

6. PLOS authors have the option to publish the peer review history of their article (what does this mean?). If published, this will include your full peer review and any attached files.

Reviewer #1: No

Reviewer #2: No

---

## [Author Response · Author response to Decision Letter 0]

23 Jan 2024

Thank you reviewers for reviewing our paper.

I have addressed all the comments raised in the paper and attached a separate document in reply to the concerns raised.

Regards

Shubhika Jain

---

## [Decision Letter · Decision Letter 1]

20 Mar 2024

Evaluating the efficacy and safety of mavacamten in hypertrophic cardiomyopathy: A systematic review and meta-analysis focusing on qualitative assessment, biomarkers, and cardiac imaging.

PONE-D-23-42379R1

Dear Dr. Jain,

We’re pleased to inform you that your manuscript has been judged scientifically suitable for publication and will be formally accepted for publication once it meets all outstanding technical requirements.

Kind regards,

Yashendra Sethi

Academic Editor

PLOS ONE

Additional Editor Comments (optional):

Congratulations on your great contribution.

Reviewers' comments:

Reviewer's Responses to Questions

**Comments to the Author**

1. If the authors have adequately addressed your comments raised in a previous round of review and you feel that this manuscript is now acceptable for publication, you may indicate that here to bypass the “Comments to the Author” section, enter your conflict of interest statement in the “Confidential to Editor” section, and submit your "Accept" recommendation.

Reviewer #1: All comments have been addressed

Reviewer #2: All comments have been addressed

2. Is the manuscript technically sound, and do the data support the conclusions?

Reviewer #1: Yes

Reviewer #2: Yes

3. Has the statistical analysis been performed appropriately and rigorously? 

Reviewer #1: Yes

Reviewer #2: Yes

4. Have the authors made all data underlying the findings in their manuscript fully available?

Reviewer #1: Yes

Reviewer #2: Yes

5. Is the manuscript presented in an intelligible fashion and written in standard English?

Reviewer #1: Yes

Reviewer #2: Yes

6. Review Comments to the Author

Reviewer #1: The manuscript is suitable for publication in its present for. The comments have been pertinently addressed

Reviewer #2: All comments have been addressed.

7. PLOS authors have the option to publish the peer review history of their article (what does this mean?). If published, this will include your full peer review and any attached files.

Reviewer #1: No

Reviewer #2: No

---

## [Editor Report · Acceptance letter]

8 Apr 2024

PONE-D-23-42379R1 

PLOS ONE

Dear Dr. Jain, 

I'm pleased to inform you that your manuscript has been deemed suitable for publication in PLOS ONE. Congratulations! Your manuscript is now being handed over to our production team.

Kind regards, 

on behalf of

Dr. Yashendra Sethi 

Academic Editor

PLOS ONE